# Development of a Non-Contacting Muscular Activity Measurement System for Evaluating Knee Extensors Training in Real-Time

**DOI:** 10.3390/s22124632

**Published:** 2022-06-19

**Authors:** Zixi Gu, Shengxu Liu, Sarah Cosentino, Atsuo Takanishi

**Affiliations:** 1Faculty of Science and Engineering, Waseda University, Tokyo 1690051, Japan; ryu_seiku@ruri.waseda.jp (S.L.); sarah.cosentino@aoni.waseda.jp (S.C.); contact@takanishi.mech.waseda.ac.jp (A.T.); 2Department of Modern Mechanical Engineering, Waseda University, Tokyo 1690051, Japan; 3Humanoid Robotics Institute, Waseda University, Tokyo 1690051, Japan

**Keywords:** muscular activity estimation, knee extensors, muscle training, rehabilitation, real-time, neuron network, IMU sensor

## Abstract

To give people more specific information on the quality of their daily motion, it is necessary to continuously measure muscular activity during everyday occupations in an easy way. The traditional methods to measure muscle activity using a combination of surface electromyography (sEMG) sensors and optical motion capture system are expensive and not suitable for non-technical users and unstructured environment. For this reason, in our group we are researching methods to estimate leg muscle activity using non-contact wearable sensors, improving ease of movement and system usability. In a previous study, we developed a method to estimate muscle activity via only a single inertial measurement unit (IMU) on the shank. In this study, we describe a method to estimate muscle activity during walking via two IMU sensors, using an original sensing system and specifically developed estimation algorithms based on ANN techniques. The muscle activity estimation results, estimated by the proposed algorithm after optimization, showed a relatively high estimation accuracy with a correlation efficient of *R*^2^ = 0.48 and a standard deviation STD = 0.10, with a total system average delay of 192 ms. As the average interval between different gait phases in human gait is 250–1000 ms, a 192 ms delay is still acceptable for daily walking requirements. For this reason, compared with the previous study, the newly proposed system presents a higher accuracy and is better suitable for real-time leg muscle activity estimation during walking.

## 1. Introduction

Knee extensors play an essential role in human mobility [1]. In fact, knee extensors are crucial for activities of daily living (ADL), such as climbing stairs [2], standing up from a chair [3]; and play a fundamental role in many sports, such as volleyball [4], cycling, and basketball [5]. As with every muscle, the knee extensors need continuous training to maintain their strength. However, since they are activated only during very specific movements, it is difficult to estimate the training effect of ADL without any feedback devices. According to the U.S. National Health and Nutrition Examination Survey (NHANES), 45% of US adults self-reported that their training is sufficient, while less than 5% of them actually met the national physical activity recommendation when measured with quantitative analysis using various sensors [6]. Therefore, it was evident that quantitative muscle activity measurement during training can be useful to help people achieve their target by avoiding insufficient or excessive training.

In the past few years, the development of wearable measurement systems improved fast because people started paying more attention to their health status and had begun monitoring their own daily activity [7,8,9]. As walking is a universal, very basic activity for humans, it became a very popular baseline for everyday overall training estimation. At the present, many wearable devices, such as wristwatches and ankle bands, can detect the number of steps of a user during walking. However, the number of steps can offer only limited information about the quality of walking, and on the activation of specific muscular groups in the legs. For a more detailed evaluation of the training effect of walking on single muscle groups, gait analysis should be performed using real-time muscular activity data.

In medical settings, muscular activity is usually assessed using surface electromyography (sEMG) [10,11,12]. The clinical usage of sEMG sensors is often referred to as kinesiological sEMG as it involves dynamic recording of muscular activity during intentional movements in functionally relevant tasks characterized by cyclic motion of limb segments such as walking and stepping [13,14,15]. In most gait analysis tests surface myoelectric signals are collected by about 10 sets of electrodes placed in a bipolar configuration connected to a preamplifier fitted a few centimeters away from the contact points. A central board is used to collect the synchronized data from all the sEMG sensors by wire cables or wireless transmission and to transfer the data to the receiving data processing unit. Muscle activity is then calculated from these surface myoelectric data. In more recent studies on gait analysis, optical motion capture systems have also been used for model-based estimation of the muscle activity [16]. Subjects were fitted with markers on all major joints of lower limbs and walked inside a room equipped with an optical motion measurement system which recorded the motions of all markers. Gait was then reconstructed using a kinematic model of the lower human body, and leg muscle activity was then estimated from the reconstructed gait. Instead of motion capture systems, multiple inertial measurement units (IMUs) have also been used in previous studies on gait analysis [17] and rehabilitation [18]. An IMU can measure motion parameters, such as acceleration and angular velocity. To estimate the human lower body link model one IMU must be placed on every link of the lower limbs kinematic chain [19]. The accuracy of IMU-based systems is comparable to the accuracy of motion capture systems [20], albeit their usability is higher as they do not require a fitted environment.

However, all these methods for measuring leg muscle activity have severe limitations, which limit their suitability for widespread adoption as muscle activity estimation systems during ADL. sEMG sensors are invasive as they require adequate skin preparation, such as shaving and removing the sebaceous film by alcohol, to ensure good electrode contact; and the signal processing phase is critical to remove noise and interferences which can corrupt the signal and lead to calculation errors [21]. Optical motion capture systems are expensive and require a specifically fitted room, which severely limits the workspace dimension and the situations in which they can be used. Multiple IMUs acquisition systems also require a long preparation time to fit all sensors in place and can grow very expensive due to the number of units to be used.

For this reason, in a past study, we have developed a system for kinematic model-based leg muscular activity estimation using only one IMU sensor placed on the shank. However, the preliminary results of system testing showed that knee extensors activity estimation using a traditional human lower limb kinematic model was not sufficiently accurate for real-time analysis.

In this study, we present an updated measurement system and muscle activity estimation algorithm that overcomes all the limitations previously mentioned. The sensor system uses two IMUs, one on the shank and one on the knee joint. The IMUs collect motion data for the calculation of four gait parameters includes acceleration, velocity, angular velocity, and stride length to reconstruct a more precise gait model. ANN techniques, which are often used to solve predictive system problems [22,23], are embedded in the data processing algorithm, using gait parameters and individual biological data, such as BMI and the ratio between lower limb joints, to estimate the activity of the knee extensors. The muscle activity measured from sEMG sensors was used as ground truth at the training phase. We also present the results of a gait analysis experiment to evaluate both performance and reliability of this new system for estimation of knee extensors activity during walking.

This paper is structured as following: Section 2 presents the materials and methods and introduces the system used. The experimental setup and algorithm are also described here. Section 3 presents results and Section 4 discusses the results and presents the future research needed to overcome the present problems.

## 2. Materials and Methods

The system overview is shown in Figure 1. The motion data collected from IMU sensors were used to calculate gait parameters. These gait parameters, together with muscular activity measured from sEMG sensors and body data were applied as input of the machine learning in the training phase. Once the training is finished, the model can estimate the muscular activity with gait parameters and body data.

### 2.1. Sensor System

The sensor system consisted of sEMG sensors and IMU sensors, both designed by our group. In particular, data from sEMG sensors were used only for neural network training and for the final algorithm validation, to compare the estimated muscle activity with the actual muscle activity directly measured via sEMG.

The sEMG sensor, named Waseda Bioinstrumentation sEMG (WB-EMMG) (Figure 2a) [24], was developed based on ISEK [25] and SENIAM standards [26]. It has a good Signal Noise Rejection (SNR) thanks to its gold plated active electrodes [27], and an appropriate AD coupling and common mode noise rejection by design [24]. The specifications of the sEMG sensor are summarized in Table 1.

The 9-axis IMU, named Waseda Bioinstrumentation Ver.4 IMU (WB-4 IMU) (Figure 2b) [28] includes a tri-axial accelerometer, a tri-axial gyroscope, and a tri-axial magnetometer. It also has both a Bluetooth interface, which allows wireless data transmission for a maximum distance of 10 m, and a CAN bus interface, which allows synchronized data transfer via cable. The IMU specifications are summarized in Table 2.

A diagram of the whole sensor system used in this study is shown in Figure 3. A Central board gathered all synchronized sEMG sensor data through CAN bus and sent it to a data processor through class1 Bluetooth interface, while IMUs sent motion data through class1 Bluetooth interface simultaneously. The data processor collected all synchronized sensors data and processed them in real-time.

### 2.2. Experimental Protocol

In this study, 14 healthy subjects (8 males, 6 females, age: 21–41 years old) participated in the data collection experiment. Subjects’ relevant physical parameters are shown in Table 3. All subjects gave their informed consent for inclusion before they participated in the study. The study was conducted in accordance with the Declaration of Helsinki, and the protocol was approved by the Ethics Committee of Waseda University (Protocol code: 2015-050 and Date of approval: 3 July 2015).

The experiment protocol was composed of 5 phases as follows:T1: 5 m normal flat walking at self-pace for five timesT2: 5 m normal flat walking at brisk pace for five timesT3: MVC of knee extension muscle (Left)T4: MVC of knee extension muscle (Right)T5: Measurement of lower limb ratio and BMI

Phases T1 and T2 were conducted on a flat surface indoors (Figure 4). Gait data from both EMG and IMUs were recorded while the subjects walked normally, for 5 m, five times, at the same speed for each phase. In phase T1, subjects walked at a self-perceived normal pace, and in phase T2, at a self-perceived brisk pace. At the beginning and the end of each phase, and every time a 5 m walking distance repetition was completed, the subjects stood without moving for 3 s. The subjects wore their normal shoes during the experiment.

After the walking experiment, in phases T3 and T4 we measured the maximum voluntary contraction (MVC) of calf and thigh of subjects. In T3 (Figure 5a), subjects laid down and the maximum muscle activation of the thigh muscles on isometric contraction was measured via EMG sensors for 5 s. Subjects were asked to push their foot as much as possible with knees bent at 90° while the shank was restrained by the experimenters. In T4 (Figure 5b), the subject sat on a chair and the maximum muscle activation of the calf muscles on isometric contraction was measured via EMG sensors for 5 s. Subjects were asked to push the anterior part of their foot against the wall as much as possible while the chair was kept in place by the experimenters.

Finally, in T5 the body measurements, weight, height, length of the calf, and length of the full leg, were taken to calculate the lower limb ratio and BMI. These parameters were calculated using the following formulas:(1)$$BMI\left(Body\;Mass\;Index\right)=\frac{Weight\left[\mathrm{kg}\right]}{Height^{2}\left[m^{2}\right]}$$
(2)$$R_{ll}\left(Lower\;limb\;ratio\right)=\frac{Lenght\;of\;the\;calf\left[\mathrm{cm}\right]}{Lenght\;of\;the\;lower\;limb\left[\mathrm{cm}\right]}$$

The target knee extensor muscles in this study were the rectus femoris muscle (RFM) in the thigh and the outer gastrocnemius muscle (OGM) in the calf. For this reason, the sensors were placed as in Figure 6. below: the two EMG sensors were placed at 1/2 of the thigh length and at 1/3 of the calf length from the knee. There are individual variations in the actual conformation of the muscles, therefore, extra care was taken in ensuring an accurate placement of the sensor on the middle of the target muscles to minimize measurement errors.

IMU sensors were positioned at the knee and the ankle, oriented so that the sensor *Z*-axis was perpendicular to the facing direction of the subjects. Unlike the EMG sensors, IMU sensors do not require direct contact with the skin. After positioning the sensors, the whole system was turned on and connected to the GUI on the data processor PC via Bluetooth.

### 2.3. Muscle Activity Estimation

The muscle activity estimation algorithm was divided into two main parts. A block diagram of the algorithm is shown in Figure 7.

#### 2.3.1. Gait Parameter Calculation

To estimate the leg muscles activity from the motion data measured via IMU, gait parameters must be calculated first. A block diagram of the gait parameters calculation algorithm is shown in Figure 8.

The gait parameters calculation algorithm consists of the following steps:Swing phase and stance phase detection: the angular velocity is used to determine the initial and terminal contact of the foot using the algorithm described in [29]. The gait cycle is then determined from the detection of the following events:Heel Strike (HS): the instant in which the foot heel touches the soil. It is the start point of the stance phase and the end point of the swing phase;Foot Flat (FF): the instant in which the foot lays flat on the soil;Toe Off (TF): the instant in which the toe leaves the soil after the foot is thrusted backward. It is both the end point of the stance phase and the start point of the swing phase;Mid Swing (MS): The instant in the middle of the period in which the foot is in the air.

An example of IMU gait data for one foot and gait parameter computation is shown in Figure 9.

The stance phase is defined as the period from HS to TO while the swing phase is defined as the period from TO to the subsequent HS. The stance time and the swing time can then be calculated.

2.Peak detection in swing phase: The peak values of acceleration and angular velocity vectors during swing phase are calculated using the following formulas:
(3)$$a_{peak}=\underset{1\leq t\leq N}{\max}∥\vec{a}(t)∥$$
(4)$$\omega_{peak}=\underset{1\leq t\leq N}{\max}∥\vec{\omega }(t)∥$$

Notably, an R-adaptive extended Kalman filter (EKF) was applied to the sensor data to increase the computational accuracy [30]. R is the covariance matrix of the acceleration, corrected to remove the effect of gravity [31]. In fact, to convert the coordinate system of the inertial sensor from local to global coordinates, the angle of rotation of the sensor in global coordinates must be calculated. The angle of rotation can be calculated from acceleration and angular velocity data. However, the angle obtained by integrating angular velocity accumulates integration error, while the value obtained by calculating the change of gravity direction from accelerometer data is affected by the acceleration error due to inertial force. Therefore, the EKF plays a fundamental role in increasing the accuracy of the whole sensor system against errors due to sensor data noise. In our case, the EKF uses accelerometer and gyroscope data for noise canceling and measurement accuracy improvement, as shown in the block diagram in Figure 10.

In the prediction phase, the angle in the following instant in time is calculated by the current angle and angular velocity. In the update phase, the predicted angle is compensated by the angle calculated from the actual accelerometer data. The accuracy of the angle calculated from accelerometer data depends on the measurement condition, whether static or dynamic. To overcome this problem, the R-adaptive process minimizes the uncertainty in each measurement condition.

The formula for computing R is shown in Equations (5) and (6) [32]:(5)$$R=\left[\begin{array}{ccc} \sigma_{x}^{2} & 0 & 0\\ 0 & \sigma_{y}^{2} & 0\\ 0 & 0 & \sigma_{z}^{2} \end{array}\right]$$
(6)$$\sigma_{k}^{2}=\frac{1}{N+1}\overset{k}{\underset{i=k-N}{\sum }}{\left(\left|\left|a_{i}\right|\right|-1\right)}^{2}$$

R-adaptive is the acceleration norm minus gravity where *N* is the number of samples of the temporal window; $\sigma_{k}^{2}$ is the estimated variance at step *k*; $\left|\left|a_{i}\right|\right|$ is the module of the acceleration measured by the accelerometer (in G) at the step i. Contrary to the accelerometer, the uncertainty of gyroscope data does not depend on the measurement condition, so it is assumed to be equal to the variance of measurement noise.

3.Coordinate transformation: To calculate the actual motion of the ankle, the motion data measured by the IMU sensor in its local coordinate must be transformed into the global coordinate as shown in Figure 11. To do so, we used the quaternion $\hat{q}$ calculated by EKF, which gives the attitude of the sensor in the 3D space.

4.Zero Velocity Update: To cancel the integration error, we used the Zero Velocity Update (ZUPT) [33] algorithm to calculate the peak of the velocity and estimate the stride length. The peak of the velocity was calculated using the following set of equations:


(7)
$$v_{x}\left(t\right)=\sum_{MS}^{HS}a_{x}\left(t\right),$$



(8)
$$v_{y}\left(t\right)=\sum_{MS}^{HS}a_{y}\left(t\right),$$



(9)
$$∥\vec{v}(t)∥=\{\begin{array}{cc} \sqrt{v_{x}{\left(t\right)}^{2}+v_{y}{\left(t\right)}^{2}} & \left(MS_{prev}<t<HS\right)\\ 0 & \left(HS\leq t\leq MS_{next}\right) \end{array}$$



(10)
$$v_{peak}=\underset{1\leq t<N}{\max}∥\vec{v}(t)∥$$


5.Stride length calculation: the stride length was calculated from the walking distance in global coordinated, which is computed by integrating the velocity in time (Equation (11)). The walking distance computed using data from the sensor on the shank is shown in Figure 12.
(11)$$X\left(t\right)=\int^{\;}v\left(t\right)dt$$

Since the distance in global coordinates traveled by the sensor at the ankle is the distance traveled by the ankle itself, the sensor is stationary during the stance phase and moves forward during the swing phase. Therefore, the graph in Figure 11 has a staircase shape. The stride length can be calculated from the walking distance via Equation (12). To improve the algorithm accuracy against missteps, strides with a stride length of 30 cm or less are ignored [34].
(12)$$l_{stride}\left(i\right)=\{\begin{array}{c} X\left(t_{HS}\left(i\right)\right)\\ X\left(t_{HS}\left(i\right)\right)-X\left(t_{HS}\left(i-1\right)\right) \end{array}\;\;\;\;\begin{array}{c} \left(i=1\right)\\ \left(i>1\right) \end{array}\;i=\{\begin{array}{c} i+1\\ i \end{array}\;\;\;\begin{array}{c} \left(l_{stride}\left(i\right)>30\left[\mathrm{cm}\right]\right)\\ \left(l_{stride}\left(i\right)\leq 30\left[\mathrm{cm}\right]\right) \end{array}$$

#### 2.3.2. Muscle Activity Measurement

In this study, we used machine learning techniques to build an adaptive model of muscle activity. In particular, we used an Artificial Neural Network (ANN) for model training. There are in fact many works using ANN for human analysis and adaptive modeling. In the initial stage of model training, the model was trained within a full-connected network. The problems of the trained model were recorded and analyzed after obtaining the estimation results, and the model was refined. After a few refining iterations, having reached satisfactory training results, the model hyper-parameters were optimized. The network structure was also modified to improve the estimation accuracy. The ratio between training datasets and validation datasets is 8:2, while the validation datasets were selected by assigned random seed in algorithm.

Since the problems discussed in this study are value prediction problems, we are essentially facing a standard regression problem for classification. The model training in this study was conducted locally. The training framework and the software environments are shown in Table 4.

The CUDA library is a parallel computing library that allows software to use certain types of graphics processing units (GPUs) for general purpose processing. It is applied in the project to get acceleration in machine learning. Tensorflow is a free and open-source platform for machine learning. Keras is also an open-source library that acts as an interface for the Tensorflow to enable fast experimentation with deep neural networks.

In the training phase, we used normalized MVC data $R_{mvc}$ instead of raw muscle activity EMG data. MVC normalization is a commonly used amplitude post-processing technique on EMG signals. The method utilizes a maximum root mean square (RMS) value from the recording of the MVC of the muscle to normalize subsequent EMG data series. The output is displayed as a percentage of the MVC (%MVC) value, which can be used to easily establish a common ground when comparing data between subjects and between different muscles. The $R_{mvc}$ calculated is used for neural network regression in order to find the correlation between gait parameters and muscle activity. Figure 13 shows the overall EMG data processing workflow.

From the EMG of one gait cycle the Root Mean Square (RMS) with 100 ms of window size (Equation (13)) is calculated, and subsequently normalized by MVC.
(13)$$RMS=\sqrt{\frac{1}{2\cdot t_{window}}\overset{t+t_{window}}{\underset{t-t_{window}}{\sum }}{\left(EMG_{Filtered}\left(t\right)\right)}^{2}}$$

## 3. Results

In this study, a new method to estimate muscle activity of two muscles used in walking, the rectus femoris muscle in the thigh (RFM), and the outer gastrocnemius muscle in the calf (OGM) is presented. The real-time EMG muscle activity measurement was taken as the ground truth against which the model performance was evaluated. We compared the performance of the muscle activity estimation model with the actual muscle activity measured via EMG during walking. We used the coefficient of determination $R^{2}$, calculated in Equation (14), as the index to evaluate the performance of the model:(14)$$R^{2}=1-\frac{SS_{res}}{SS_{tot}}=1-\frac{\sum_{1}^{n}{\left(y_{ESTIMATION}-y_{REAL}\right)}^{2}}{\sum_{1}^{n}{\left(y_{ESTIMATION}-y_{AVG}\right)}^{2}}$$
where $SS_{res}$ refers to the residual sum of squares, and $SS_{tot}$ refers to the total sum of squares.

For each muscle, the average muscle activity and the peak performance in one gait cycle were estimated. The results of the system performance experiment are shown in Table 5. The comparison between the muscle activity estimated by the proposed algorithm and the actual % MVC calculated from EMG measurements showed a high estimation accuracy with a correlation efficient of $R^{2}$ = 0.89.

The real-time performance was shown as Table 6.

## 4. Discussion

By default, the hyper-parameters of the model at this stage have not been optimized. From the views of results, the accuracy of the model is notably high, but weird where accuracy testing is 30% different between the training dataset and the testing dataset. Such a situation shows that the model is with low robustness at the same time. We achieved the model optimization to improve the robustness.

Optimization of the Hyper-parameters: Hyper-parameters refer to the parameters that control the process of machine learning: loss function, learning rate, and optimization algorithm. The model optimization used in this study is “GrindSearch” [35]. This method is simply an exhaustive searching through a specified subset of hyper-parameters of the learning algorithm to find the optimal combination. This method has two main limitations. First, this hyper-parameter optimization method is a brute force search method. Before using it, it is necessary to set a limited range for each hyper-parameter, and the obtained optimal result is only the best result in the set range. Second, because this method will exhaustively compare the performances of the algorithm for all the hyperparameters combinations within the given range, it is very time-consuming. Even in the limited hyperparameters subspace used in this study, it took a long time.The subset of hyperparameters optimized in this study is:(a)the number of hidden layers;(b)the number of nodes in the hidden layer;(c)the activation function of hidden layers;(d)the optimizer;(e)the loss function.

For (number of the training = 3) in the given range combination, the final selected hyperparameter values are shown in Table 7.

This set of optimized hyperparameters leads to a relatively good performance, but also to a more complex structure of the model, which indirectly increases the difficulty of model training. For this reason, the optimized model is not suitable for online learning predictions. However, the model discussed in this study is only used for offline learning.

2.Optimization of the network structure: the most common network structure for ANN is full connection, meaning that the nodes of each layer are connected to all the nodes of the adjacent layers. The advantage of such a structure is that it can learn the training data characteristics more comprehensively. However, sometimes, a full connected ANN might give too much weight to less important or interfering features, e.g., signal processing noise. In such cases, the resulting trained model is less robust, e.g., overly sensitive to noise, and easy to overfit if trained on a small dataset, which leads to lack of practical application value of the model.

Therefore, in this study we used the “Dropout” method, to randomly discard some nodes connections (Figure 14). This process is equivalent to noise reduction. Before using the "Dropout" method, we only need to specify the percentage of node connections required to be dropped. We compared the performances of different neural networks obtained with this process separately and selected the structure with the best performance. In this study, 20% of the original full connected neural network connections were randomly discarded based on empirical values.

The performance gap between the model on the training dataset and the test dataset is within 5%. In other words, after this optimization process, the general performance of the model has been improved.

The results estimated by the proposed algorithm after optimization (in Table 8 and Figure 15) showed a relatively high estimation accuracy with a correlation efficient of $R^{2}$ = 0.48 (where significant difference between distributions was confirmed using the Steel-Dwass test). For the RFM, the muscle average activity estimation and the activity peak value estimation are comparable in terms of accuracy, while the first is more scattered compared to the latter. As for the OGM, the muscle average activity estimation and the activity peak value estimation are concentrated, while the first is slightly higher than the latter.

It is important to note that the performance of real-time muscle activity estimation is affected by the signal transmission and processing delay of the current system. The measured average processing delay of the system is 72 ms, while the wireless communication average delay is 120 ms, with a total system average delay of 192 ms. Human regular walking frequency is between 1 Hz and 4 Hz [36], which means that the average interval between different gait phases is 250–1000 ms. For this reason, a 192 ms delay is still acceptable for daily walking requirements (total delay <250 ms).

From the results, $R^{2}$ is near 0.5 which shows to be moderate effect size. We can say that the muscle activity estimation using proposed system is working while the estimation of the rectus femoris muscle is less accurate than the muscle activity estimation of the outer gastrocnemius muscle. The reason might be due to the scarce usage of the thigh muscle during normal walking. The function of the rectus femoris during walking is raising the leg during the swing phase and lifting the upper body during stance phase. During normal walking, the upper body will only be lifted slightly. According to a previous study [37], the activity of the rectus femoris muscle is about 26%MVC during normal walking while it is 67% MVC during stair climbing. This might also cause an error in the muscle average activity estimation, and this could explain why the activity peak value estimation of the rectus femoris muscle has better performance in comparison. As for the outer gastrocnemius muscle, both estimation results are good, and the average activity estimation is better. This is because the outer gastrocnemius muscle is frequently used in normal walking to maintain the balance of the whole body and for kicking off the ground during the stance phase.

During this study, some limitations have been discovered:The model used in the study uses the EMG sensors measurements as the ground truth for MVC calculation. However, EMG measurement itself has intrinsic errors. For this reason, further improvements in prediction accuracy are required, possibly using more than one method to measure muscle activation and calculate the MVC;The system in this study has data processing limitations. Before importing the model, the measured data must be filtered to remove noise and interference. As a result, the system must rely on adequate data preprocessing for correct real-time estimation of MVC values.

## 5. Conclusions and Future Works

In this study, a real-time muscular activity measurement system using non-contact sensors was developed. The system uses two IMU sensors to collect the motion data during normal walking to estimate the knee extensor activity. An Artificial Neural Network was used in estimation model training. An evaluation experiment was also conducted to validate the estimation algorithm.

In this study, we have shown that it is possible to estimate knee extensors activity during simple walking at different speeds, without an EMG or a Ground Reaction Force (GRF) sensor. The results of this study reveal the possibility of muscular activity estimation only by IMU.

Future works will mainly focus on the following three aspects:Improving the accuracy of the model: subsequent research will consider using multiple sensors to estimate the MVC reference value rather than a single EMG sensor to provide the MVC reference value in model training. The fusion of multiple sensors can complement the observation results, and it is expected to raise the upper limit of prediction accuracy due to the EMG sensor;Optimizing the effect of real-time prediction: subsequent research will consider simplifying the complexity of calculations and seeking more efficient algorithms, using feature extraction methods without gait cycle estimation, if it is possible;Working on applications: the proposed system can not only work as measurement to offer feedback to human, but also send muscle activity data to a smart training device or exoskeleton. We have developed a knee extensor training device with active components to apply load on the knee joint during normal walking. With more detailed, real-time muscle activity data, the device can adjust its load according to the current muscle activation to avoid insufficient or excessive training.

## Figures and Tables

**Figure 1 sensors-22-04632-f001:**
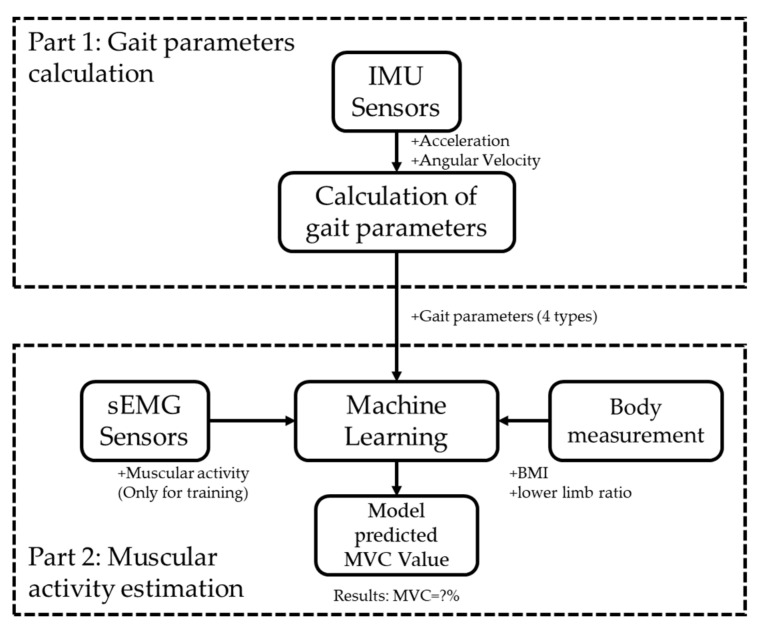
The overview of the system.

**Figure 2 sensors-22-04632-f002:**
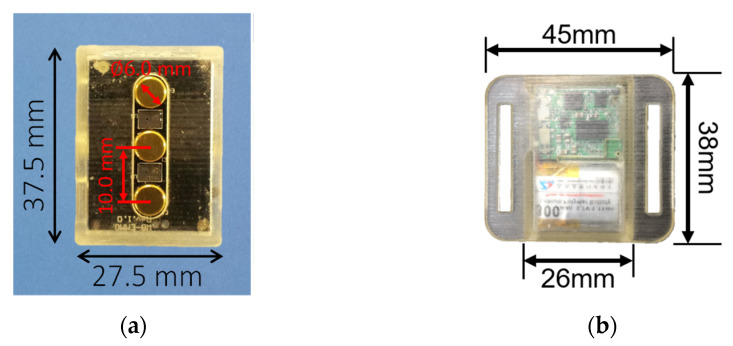
WB sensors: (**a**) WB-EMMG sensor and (**b**) WB-4 IMU sensor.

**Figure 3 sensors-22-04632-f003:**
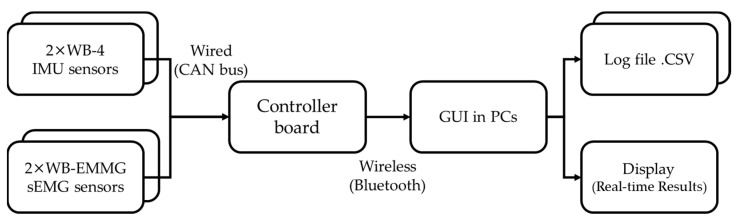
Measurement system using in the experiment.

**Figure 4 sensors-22-04632-f004:**
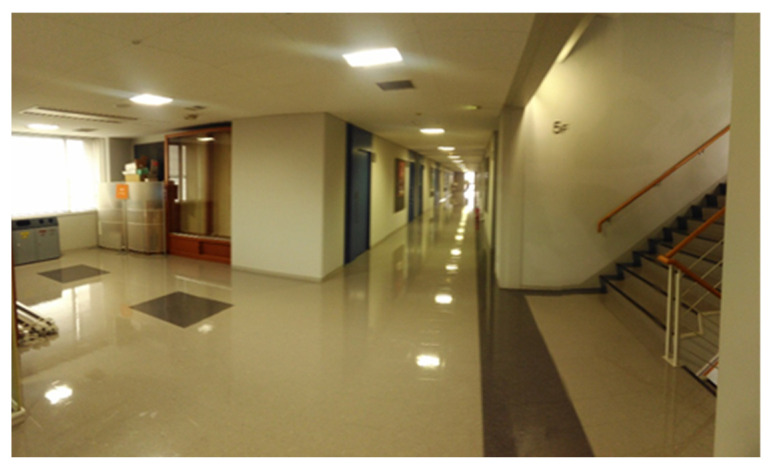
The experiment environment.

**Figure 5 sensors-22-04632-f005:**
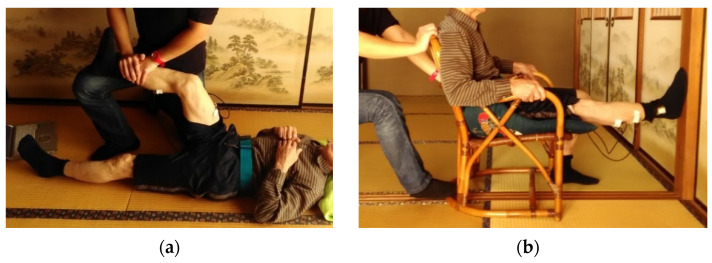
MVC measurement method: (**a**) measurement for the thigh (**b**) measurement for the calf.

**Figure 6 sensors-22-04632-f006:**
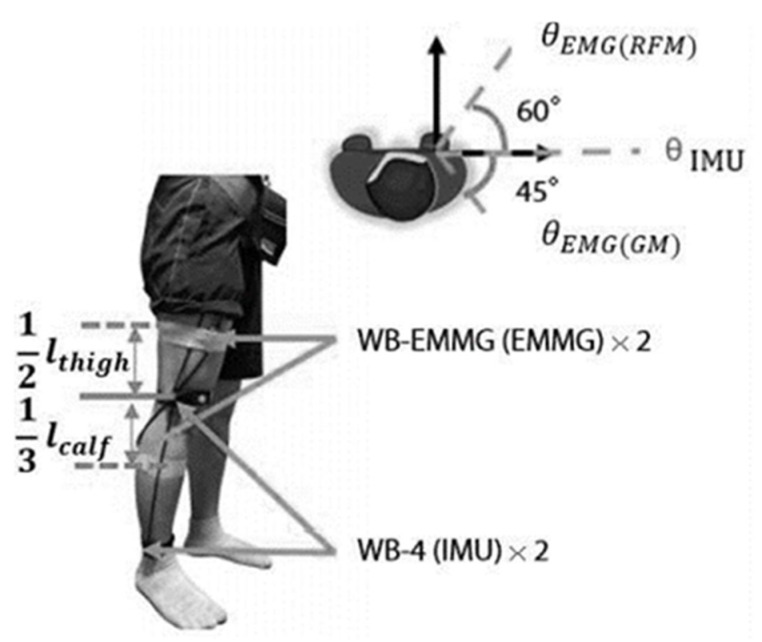
Position of IMU & sEMG sensors setup in the experiment.

**Figure 7 sensors-22-04632-f007:**
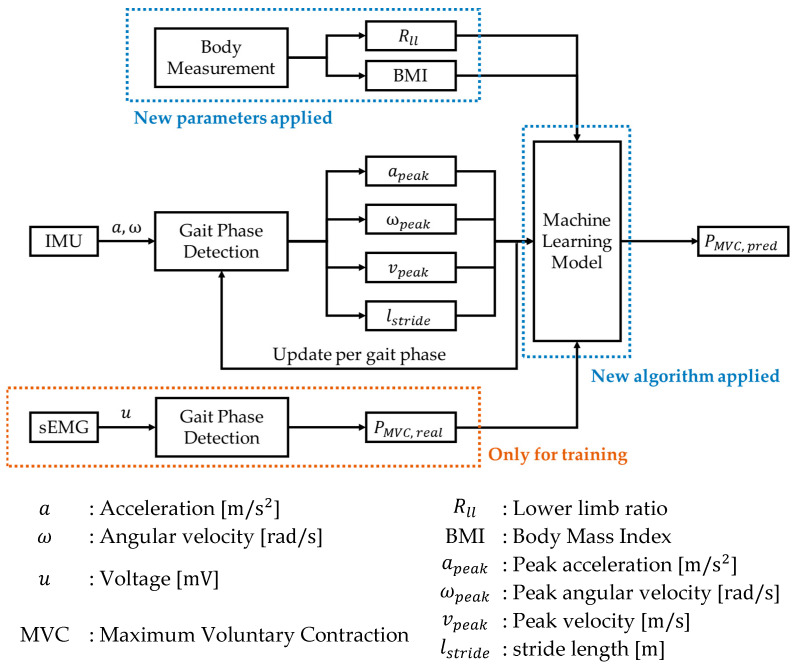
Overview of muscle activity estimation system.

**Figure 8 sensors-22-04632-f008:**
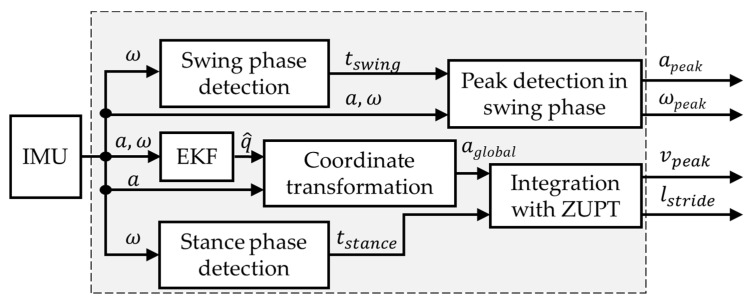
The calculation algorithm of gait parameters.

**Figure 9 sensors-22-04632-f009:**
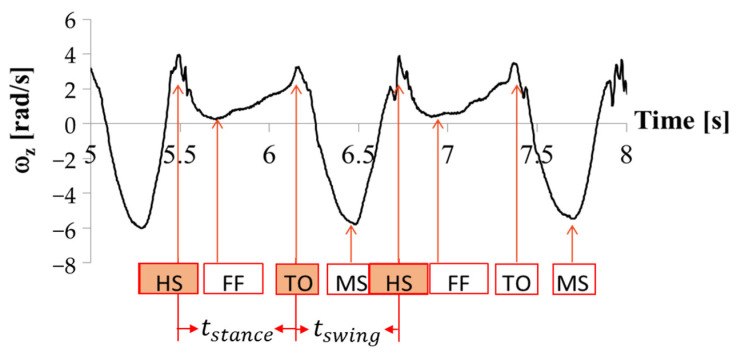
The motion data example of angular velocity for one foot.

**Figure 10 sensors-22-04632-f010:**
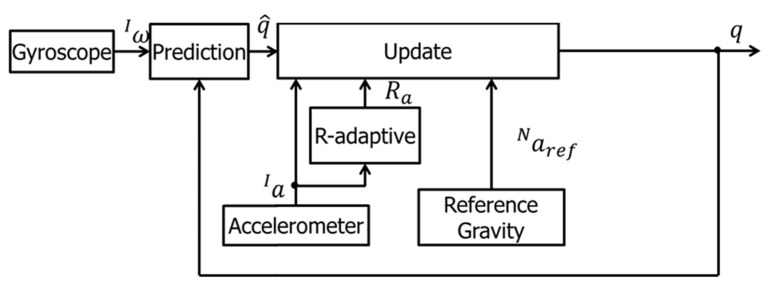
EKF process flowchart.

**Figure 11 sensors-22-04632-f011:**
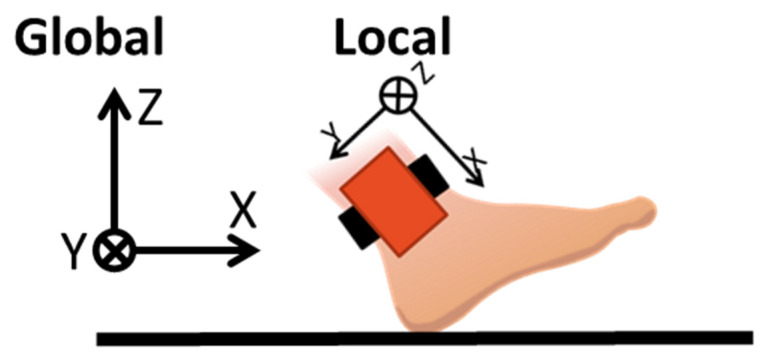
Walking distance of IMU on the ankle.

**Figure 12 sensors-22-04632-f012:**
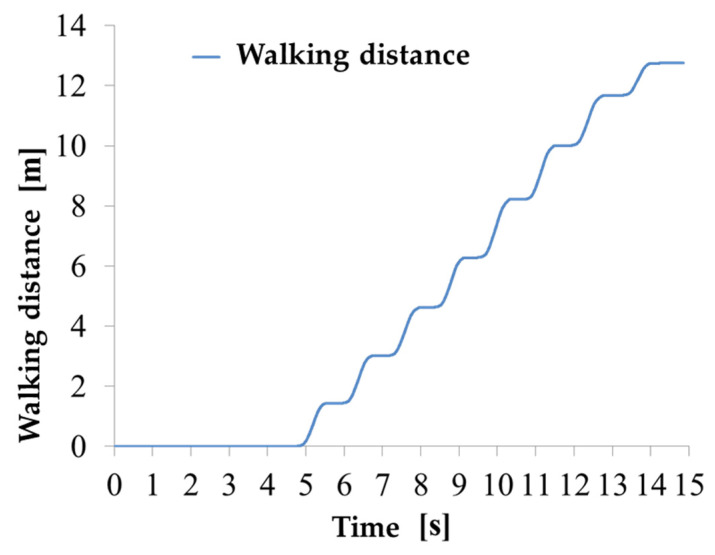
Walking distance of IMU on the ankle.

**Figure 13 sensors-22-04632-f013:**
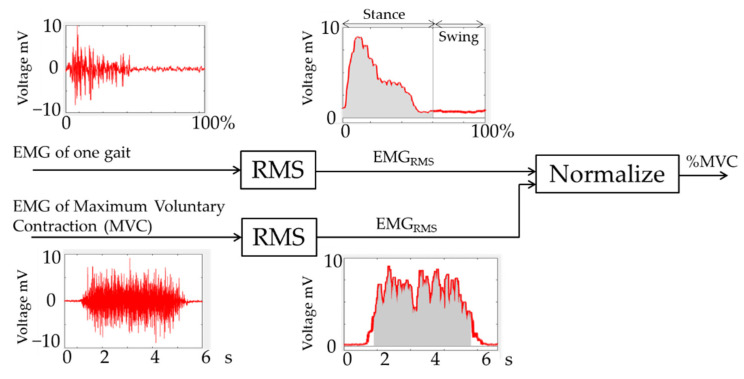
EMG data processing workflow.

**Figure 14 sensors-22-04632-f014:**
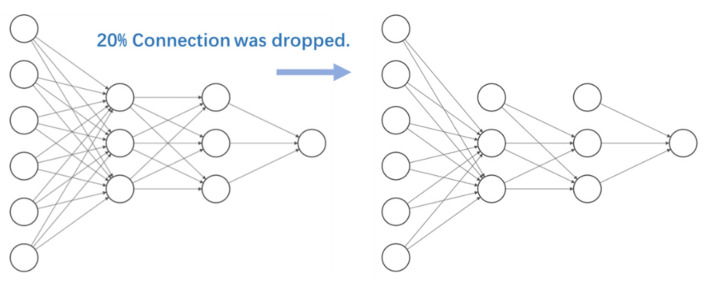
Dropout method, randomly removing the connection between nodes.

**Figure 15 sensors-22-04632-f015:**
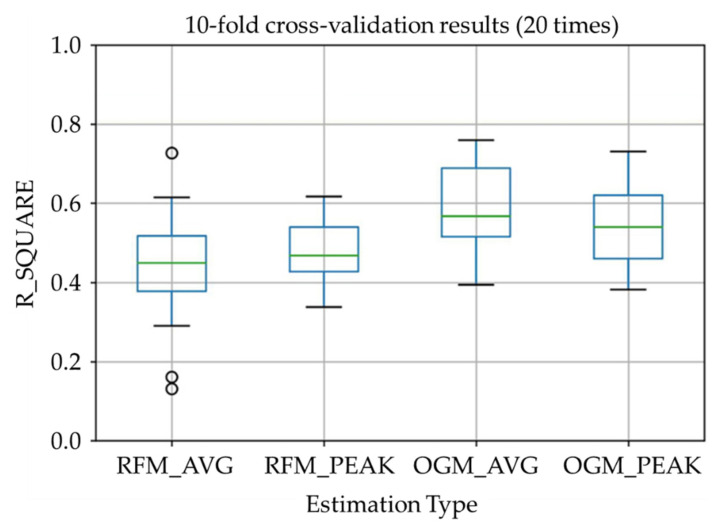
Model performance boxplot.

**Table 1 sensors-22-04632-t001:** Specification of WB-EMMG sensor used in the experiment.

Parameter	Value
Size (W × D × H) [mm]	27.5 × 37.5 × 18.0
Weight [g]	13.4
Gain	1400
Sampling rate [Hz]	1000
Resolution [Bit]	12
Lower cutoff frequency [Hz]	20
Higher cutoff frequency [Hz]	450
CMRR [dB]	>90

**Table 2 sensors-22-04632-t002:** Specification of WB-4 IMU used in the experiment.

Parameter	Value
Size (W × D × H) [mm]	45.0 × 38.0 × 12.0
Weight [g]	34
Communication Distance	Bluetooth (<10 [m])
Orientation range	360° for all axes
Resolution [degree]	<0.05°
Accuracy	<0.5° (static), <2.0° RMS (dynamic)
Sampling Rate [Hz]	400

**Table 3 sensors-22-04632-t003:** Details of the participants.

Parameter	BMI		Lower Limb Ratio	
Value	Avg.	Std.	Avg.	Std.
	21.5	3.4	0.43	0.04

**Table 4 sensors-22-04632-t004:** Local training environments.

Environments	Python	CUDA	Tensorflow	Keras
Version	3.6.5	10.0	2.0.0	2.3.1

**Table 5 sensors-22-04632-t005:** The performance of the muscular activity estimation.

	Estimation Type	$\mathrm{Estimation}\;\mathrm{Accuracy}\;(R^{2})$
MVC Value	$MVC_{AVG}\left(RFM\right)$	0.89
	$MVC_{PEAK}\left(RFM\right)$	0.26
	$MVC_{AVG}\left(OGM\right)$	0.78
	$MVC_{PEAK}\left(OGM\right)$	0.75

**Table 6 sensors-22-04632-t006:** Real-time performance (Latency).

Time Unit	Computer Processing	Wireless Transfer	Total
milliseconds	72	120	192

**Table 7 sensors-22-04632-t007:** the optimized hyper-parameters combination.

Parameters	Value
n_HiddenLayer	7
n_NeuronUnit	12
act_Func	‘relu’
Optimizer	‘Adam’
loss_Func	‘logcosh’

**Table 8 sensors-22-04632-t008:** The performance of the latest model for 20 times with model optimization (January 2022).

Estimation Type	$\mathrm{Estimation}\;\mathrm{Accuracy}\;(R^{2})$	
	Avg.	Std.
$MVC_{AVG}\left(RFM\right)$	0.48	0.10
$MVC_{PEAK}\left(RFM\right)$	0.47	0.07
$MVC_{AVG}\left(OGM\right)$	0.58	0.11
$MVC_{PEAK}\left(OGM\right)$	0.55	0.09

## Data Availability

Data presented in this study are available from corresponding authors upon request.

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
