# Peer review of "Development of a Non-Contacting Muscular Activity Measurement System for Evaluating Knee Extensors Training in Real-Time"

_sensors, 2022, doi:10.3390/s22124632_

Round 1
Reviewer 1 Report
The authors described two IMU sensors and a machine learning-based muscle activity measurement system. The study's success could lead to a system that can monitor muscular activities during everyday life. The authors had developed a system using one IMU sensor and believed two sensors would lead to better results.
The paper was technically developed and well structured. Details of the sensor system, the muscle activity estimation algorithm, and the experimental protocol were provided.
Here are my comments:
1. There is a lack of description regarding the participants. What were the inclusion/exclusion criteria for the subjects? How the subjects' data were used for training the machine learning (were some subjects' data used for training, the rest for evaluation or all of them were used to train the machine learning algorithm?)
2. Are the estimation accuracies for the individual participant or the mean of all the participants? It would be better to provide both means and standard deviations.
3. What were the walking speeds during the experiment? If the purpose of the system is to be used for everyday life, how does the delay of the system impact the accuracy during high-speed activities of daily living?
4. More discussion could be added for the application of the system based on the low accuracy.
Reviewer 2 Report
The authors present the article entitled “Development of a non-contacting muscular activity measurement system for evaluating knee extensors training in real-time”.
This paper presents an updated measurement system and muscle activity estimation algorithm. The sensor system uses two IMUs, one on the shank and one on the knee joint, to reconstruct a more precise leg kinematic model. Machine learning techniques are embedded in the data processing algorithm, using also both gait parameters as well as individual biological data like BMI and the ratio between lower limb joints to estimate the knee extensors activity.
The article presents the following concerns:
Abstract section: The Abstract focus mainly on the last work. I suggest restructuring the work by presenting the importance of the contributions of the proposed work and a pertinent overview of the work. Include quantitative value in the abstract in order to highlight the findings.
References [12-15] and [16-19]: Please mention the main contribution separately.
Line 104: Please define EMG.
Introduction section: There is a lack of information about using Machine learning techniques and the background.
It is mandatory to extend the experimental section since it is very poor.
line 258 can be justified by considering ANN fresh references as Self-tuning neural network PID with dynamic response control; Neural network and spatial model to estimate sustainable transport demand in an extensive metropolitan area; Impact of eeg parameters detecting dementia diseases: a systematic review.
Figure 12 must be placed in result section.
In general, the results and discussion section are not impressive. In its current form, it seems that only summarizes the last work. However, how was the ANN method performance? What was the final ANN structure? What were the input variables? It needs to describe in detail the results of the work. Also, in the Discussion section, describe the results and how they can be interpreted from the perspective of previous studies and of the working hypotheses.
-
The text must be written in the 3rd person or passive voice.
-
I recommend giving an introduction between sections 2 and 2.1.
-
Vectorize the images and figures.
-
Check the reference style [7] according to the instructions for the authors of the journal.
-
The literature should be updated by using up-to-date references. More than 50% of references are from more than ten years ago.
-
Aphosthrophes must be avoided
-
Vectorize the figures in order to see details. Improve the quality.
The following misspelling should be checked:
-
Line 39: “help people achieving…” should be rewritten as “ to help people achieve…”
-
Line 42: “people started pay more attention to their health status and begun…” should be rewritten as “people started paying more attention to their health status and have begun…”
-
Line 74: “shaving and remove the sebaceous film by alcohol…” should be rewritten as “shaving and removing the sebaceous film with alcohol…”
-
Lines 88-91: “Machine learning techniques are embedded in the data processing algorithm, using also both gait parameters as well as individual biological data like BMI and the ratio between lower limb joints to estimate the knee extensors activity” should be rewritten as “Machine learning techniques are embedded in the data processing algorithm, using gait parameters and individual biological data like BMI and the ratio between lower limb joints to estimate the knee extensors' activity”.
-
Lines 150-151: “Subjects were asked to push their foot as much as possible with knee bent at 90° while the shank was restraint by the experimenters” should be rewritten as “Subjects were asked to push their foot as much as possible with knees bent at 90° while the shank was restrained by the experimenters”
Round 2
Reviewer 2 Report
The manuscript has been improved greatly